# New Light on Plants and Their Chemical Compounds Used in Polish Folk Medicine to Treat Urinary Diseases

**DOI:** 10.3390/ph17040435

**Published:** 2024-03-28

**Authors:** Beata Olas, Waldemar Różański, Karina Urbańska, Natalia Sławińska, Magdalena Bryś

**Affiliations:** 1Department of General Biochemistry, Faculty of Biology and Environmental Protection, University of Łódź, Pomorska 141/143, 90-236 Lodz, Poland; natalia.slawinska@edu.uni.lodz.pl; 2Clinic of Urology and Urological Oncology, Medical University of Łódź, Copernicus Memorial Hospital, Pabianicka 62, 93-513 Lodz, Poland; walroz@onet.pl; 3Faculty of Medicine, Medical University of Łódź, 90-419 Lodz, Poland; karina.urbanska@stud.umed.lodz.pl; 4Department of Cytobiochemistry, Faculty of Biology and Environmental Protection, University of Łódź, Pomorska 141/143, 90-236 Lodz, Poland; magdalena.brys@biol.uni.lodz.pl

**Keywords:** urinary tract diseases, ethnopharmacology, ethnobotany, phytochemicals

## Abstract

This review contains the results of Polish (Central Europe) ethnomedical studies that describe the treatment of urinary tract diseases with wild and cultivated plants. The study includes only the plants that are used to treat the urinary tract, excluding prostate diseases. A review of the literature was carried out to verify the pharmacological use of the plants mentioned in the interviews. Based on this, the study reviews the pharmacological activities of all the recorded species and indicates their most important chemical compounds. Fifty-three species (belonging to 30 families) were selected for the study. The Compositae (eight species), Rosaceae (six species), and Apiaceae (six species) are the most common families used in the treatment of urinary diseases in Polish folk medicine. Both in vitro and in vivo studies have confirmed that many of these plant species have beneficial properties, such as diuretic, antihyperuricemic, antimicrobial, and anti-inflammatory activity, or the prevention of urinary stone formation. These effects are exerted through different mechanisms, for example, through the activation of bradykinin B2 receptors, inhibition of xanthine oxidase, or inhibition of Na^+^-K^+^ pump. Many plants used in folk medicine are rich in phytochemicals with proven effectiveness against urinary tract diseases, such as rutin, arbutin, or triterpene saponins.

## 1. Introduction

Urological diseases are most commonly associated with the filtration and excretion of urine from the body. In males, health problems can manifest in the urinary tract and/or the reproductive organs, while in females these issues usually affect only the urinary tract. There are many urologic disorders and diseases. They include infections of the urinary tract, urinary lithiasis, urinary incontinence and disorders of micturition, hereditary diseases, and other common urological conditions [1,2,3].

Urinary tract infections (UTIs) occur in the entire population, but young children, pregnant women, postmenopausal women, and men with prostatic hyperplasia are especially susceptible to them [1,2,3].

Urolithiasis is among the most commonly diagnosed diseases of the urinary tract. The frequency of its occurrence is about 9% of the general population [4,5]. Urolithiasis occurs in about 10% of men and 6% of women [6,7]. Over a period of five years, the risk of disease recurrence and the chance of second stone formation is 30% to 50% [8,9].

There are four categories of plants utilized in the treatment of urological diseases—botanical diuretics, urinary antiseptic and anti-adhesion agents, antinephrotoxic botanicals, and herbs used in the treatment of benign prostatic hyperplasia. Numerous plants are used as diuretics in traditional medicine. In many cases, their activity has been confirmed by preliminary clinical trials, both in healthy people and patients with urologic diseases, though the mechanism of their action often remains unclear [10]. Although these traditional herbal drugs are called diuretics, it would be more accurate to describe them as aquaretics. They usually contain flavonoids, volatile oils, saponins, or tannins, which promote blood flow in the kidneys, leading to a higher glomerular filtration rate and increased urine volume. However, they do not decrease the resorption of Na^+^ and Cl^−^ in the renal tubules; these electrolytes are retained in the body instead of being excreted with water [11]. Increased urine flow can help with the prevention of kidney stone formation [8]. Diuretic herbs can also prove useful for minor infections, which can be alleviated due to increased urine volume. Some herbs exhibit antibacterial properties, which, in combination with increased urinary output, are useful in combating infection. There are two major mechanisms of this process—targeted killing of microbes and interference in their adhesion to epithelial cells [10].

It is believed that rural, traditional folk medicine is more prevalent in underdeveloped communities, for example, in Eastern Europe (Poland, Ukraine, Lithuania, and Belarus). Traditional folk medicine and self-treatment are strongly associated with Polish culture and tradition. Between the 1940s and 1980s, it was associated with ignorance. Folk treatment methods were vigorously condemned, while folk healers were forced to abandon their practice by the “socialist health service”. However, folk treatment methods did not disappear in rural regions; ethnological, sociological, and medical studies suggest that they seem to be enjoying social acceptance [12,13,14].

The present paper collects information about plants used to treat urological diseases in Polish folk medicine and reviews their pharmacological activities, most important phytochemicals, and mechanisms of action. We described 53 species belonging to 30 plant families. The greatest number of plant species used to treat urinary diseases were found to come from the Compositae (eight species), Rosaceae (six species), and Apiaceae (six species). The plants were identified based on ethnopharmacological studies, Polish herbalist literature, and interviews with Polish traditional healers. We used PubMed, Scopus, Google Scholar, Web of Science, Science Direct, and other databases to search for the ethnomedicinal uses of these plants, as well as phytochemistry, pharmacology, toxicity, and clinical studies. The search terms included the name of plant or phytochemical and the words “diuretic”, “urinary”, “UTI”, or “urolithiasis”. Plant names have been checked and updated with the online website The Plant List version 1.1 (www.theplantlist.org (accessed between 1 April 2023 and 30 July 2023)) of the Royal Botanic Gardens, Kew, and Missouri Botanical Garden and the online website Atlas of Vascular Plants of Poland (http://www.atlas-roslin.pl (accessed between 1 April 2023 and 30 July 2023)). The common names of the plants were identified by referring to research articles.

Table 1 presents the plant nomenclature, main Polish name, English common name, the part used, utilization, and administration. Most often, dried material was used. Moreover, in Table 1, plants are listed alphabetically by plant family, and then by species.

## 2. Plants Whose Therapeutic Effect on the Urinary Tract Has Been Confirmed by Scientific Findings (in Alphabetical Order)

Based on the literature data, we found that the described plants show mainly diuretic and natriuretic effects. Table 2 presents the results of the researched studies, including the experimental model, extract used, dose, control and reference substance, duration, number of animals or participants, and the effect on UV and UNa. The remaining test results are described in the text for individual plants or are included in the table in the ‘other effects found’ column: in this way, the text complements the information in the table rather than duplicates it.

### 2.1. Achillea millefolium *L.*

The Compositae is one of the best-known families, incorporating numerous flowering plants classified into approximately 23,000 species. One such species is *Achillea*, whose members have numerous pharmaceutical properties. They are recommended as effective tonics, sedatives, and diuretics [26]. In addition, it is common to see the consumption of herbal teas from different species of *Achillea* in folk medicine. Many of the varied therapeutic usages have been confirmed by new in vitro and in vivo experimental and clinical studies. Various authors note that *Achillea millefolium* has important traditional and ethnomedicinal uses when drunk as a tea, including treating urinary disorders; it has been used in various countries, particularly in Europe [26,83,84]. The varied effects of these plants may be due to the presence of numerous secondary metabolites, i.e., flavonoids, phenolic acids, terpenoids, and sterols. Souza et al. [62] demonstrated that oral administration of *A. millefolium* extract in rats effectively increases diuresis. For example, hydroethanolic extract at the dose of 300 mg/kg increased diuresis by approximately 30–60% between 4 and 8 h after administration. This effect is dependent on the activation of bradykinin B2 receptors and the activity of cyclooxygenase.

### 2.2. Acorus calamus *L.*

For centuries, *Acorus calamus* has been used in Polish folk medicine to treat urinary diseases [85]. It is also used in modern medicine. Ghelani et al. [15] report that an ethanolic extract of *A. calamus* rhizome demonstrates diuretic and antiurolithiatic activity in an experimental animal model (male Wistar albino rats). Their results indicate that the ethanolic extract of *A. calamus* significantly increases diuresis and reduces the urinary excretion of phosphate, oxalate, calcium, and uric acid (compared to urolithiatic control rats). At the dose of 750 mg/kg, the diuretic effect was comparable to that of furosemide. The urine volume increased from 1.88 ± 0.18 mL (control) to 3.78 ± 0.11 mL (*A. calamus* extract at the dose of 750 mg/kg) and 4.02 ± 0.17 mL (furosemide at the dose of 15 mg/kg).

### 2.3. Aegopodium podagraria *L.*

*Aegopodium podagraria* is a perennial herb of the carrot family (Apiaceae). It is native to Europe, western and eastern Siberia, the Caucasus, and Central Asian mountainous regions. It has been naturalized in North America. Koyro et al. showed that the essential oil from the flowers has diuretic and uricosuric activities [63]. Another study examined the influence of the extract and tincture of the aerial parts of *A. podagraria* in rats that received excess fructose with hydrochlorothiazide [64]. The authors observed that the tested extract significantly enhances kaliuresis.

### 2.4. Amaranthus retroflexus *L.*

*Amaranthus retroflexus* is an upright annual herb that was probably introduced into Europe from North America. Currently, it is common throughout Poland [86]. The plant is known to have a number of toxic effects; for example, Osweiler et al. [87] demonstrated that *A. retroflexus* induced the production of perirenal edema in pigs. Microscopical lesions include renal tubular degeneration and necrosis. [88] observed that acute renal failure is associated with *Amaranthus* species, including *A. retroflexus* ingestion by lambs.

### 2.5. Apium graveolens *L.*

In traditional medicine, some medicinal plants, i.e., corn silk, barley, and celery were used to relieve renal pain. *Apium graveolens* is a popular vegetable that can be added to many dishes, such as salads. It has been used in Chinese medicine to decrease blood pressure and in Arabian medicine to relieve renal pain [10,65,89,90]. [65] observed that celery (at the dose of 8 g/kg/day) increased urinary Ca^2+^ excretions (from 2.05 ± 0.07 to 5.01 ± 0.05 mmol/L) in an experimental model of nephrocalcinosis, and caused a significant reduction in serum creatinine (from 97.3 ± 0.5 to 87.2 ± 0.63 mmol/L) and blood urea nitrogen (from 7.3 ± 0.2 to 5.7 ± 0.5 mmol/L). Renal functions were analyzed on the first, fifth and 10th day.

### 2.6. Arctostaphylos uva-ursi (*L.*) Spreng.

*Arctostaphylos uva-ursi* folium has been used for therapeutic purposes in Europe and America, particularly in the treatment of lower urinary tract infections. The key constituents of the dried leaves are arbutin (a phenolic glycoside) and its derivate hydroquinone, which both show antiseptic activity in inflammation of the urinary tract [36,91]. Although arbutin is the major pharmacological active constituent of the *A. uva-ursi* leaves extract, experimental studies indicate that their global pharmacological action requires the use of the whole extract. Haslam [92] described that the dried leaves of *Arctostaphylos uva-ursi* have a soothing, astringent effect and that they can be used as a diuretic in kidney disorders and aliments of the bladder and urinary tract. The principal phenolic metabolites in the leaves are arbutin, gallotannins, and galloyl esters of arbutin.

### 2.7. Betula pendula Roth and Betula pubescens Ehrh.

Although over a hundred *Betula* species are distributed all over the globe, we only know of seven species used in traditional medicine, *B. pendula* among them. Crude extracts, fractions, and phytochemical constituents isolated from *B. pendula* have demonstrated a wide spectrum of in vivo and in vitro biological properties [93]. Extracts of *B. pendula* leaves have been used in arthritic diseases and to relieve rheumatic pain. In addition, different experiments suggest that these extracts have mild diuretic activity and anti-inflammatory properties. *B. pendula* extract reduces the growth and proliferation of activated T lymphocytes in a dose-dependent manner; this has been attributed to the action of secondary metabolites (i.e., phenolic acids and flavonoids) within the leaves [94,95]. Also, an ethanol–water (1:1; *v*/*v*) extract from the *Betula* spp. leaves has been found to have an antiadhesive effect against the binding of uropathogenic *Escherichia coli* to the bladder cell surface (cell line T24). Decreased bacterial adhesion (IC_50_ 415 mg/mL ± 7.19) was observed, and this action is linked with urinary tract infection prevention [66].

### 2.8. Bidens tripartita *L.*

Another member of the Composite is *Bidens tripartita*. It is also known in oriental medicine, where it is used as a diuretic and diaphoretic in nephrolithiasis. It was also valued for its antiseptic and anti-inflammatory properties [27,28,96].

### 2.9. Carum carvi *L.*

The dried fruits of *Carum carvi* are used in traditional medicine as a carminative and have been found to be effective against gastrointestinal complications [19]. In Moroccan traditional medicine, the ripe fruits of *C. carvi* are used as diuretics; however, this activity has not been investigated in controlled studies. On the other hand, Lahlou et al. [68] report that the aqueous extract of *C. carvi* demonstrated diuretic properties in healthy rats.

### 2.10. Daucus carota *L.*

*Daucus carota* commonly known as carrot, has been used traditionally in folk medicine to treat nephrosis and other urinary disorders. Sodimbaku et al. [69] have observed that carrot has dose-dependent nephroprotective activity against gentamicin-induced nephrotoxicity in rats. Both used doses (200 and 400 mg/kg) had significant effects. The dose of 400 mg/kg decreased the levels of serum urea (from 79.10 ± 2.21 to 45.96 ± 2.37 mg/dL), blood urea nitrogen (from 37.23 ± 1.36 to 20.21 ± 3.38 mg/dL), uric acid (from 4.87 ± 0.29 to 2.83 ± 0.35 mg/dL), and creatinine (from 2.03 ± 0.08 to 1.04 ± 0.08 mg/dL). Moreover, concurrent treatment with *D. carota* extract significantly inhibited gentamicin-induced weight loss.

### 2.11. Elsholtzia ciliata (Thunb.) Hyl.

*Elsholtzia ciliata* is widely distributed throughout Korea, China, and Europe. In Poland, it grows mainly in the eastern regions [97,98]. An in vivo study found that ethanol extract from this plant inhibits renal interstitial fibrosis induced by unilateral ureteral obstruction. This effect may be mediated by inhibiting the expression of KIM1 (kidney injury molecule 1), TNF-α (tumor necrosis factor α), TGF-β (transforming growth factor β), Smad3, and MMP 9 (matrix metalloproteinase 9) proteins, which are markers of inflammation and renal histopathological alterations. An in vitro study based on MTT (3-(4,5-methylthiazol-2-yl)-2,5-diphenyltetrazolium bromide) assay showed that *E. ciliata* ethanolic extract has no cytotoxic effect, even at a concentration of 200 µg/mL. The study was conducted for 24 h on mouse macrophage cells (RAW 264.7) and human renal mesangial cells [70].

### 2.12. Elymus repens (*L.*) Gould

*Elymus repens* originates from the temperate regions of Europe and Central Asia. Nowadays, it can be found in Africa as well. Traditionally, it is used as a soothing diuretic and to alleviate pain and spasming of the urinary tract. It is often used in urinary system disorders in children (e.g., urinary incontinence and enuresis), to combat the symptoms of urinary disease, urinary calculi, and urinary infections (like prostatitis, urethritis, or cystitis) [99]. In a clinical study, the effects of administrating *A. repens* with potassium citrate were assessed in patients with nephrolithiasis. After five months the treated group exhibited a significant reduction in the diameter (−3.6 ± 0.9 mm vs. 0.0 ± 0.8 mm) and total number of stones (−1.0 ± 0.2 vs. 0.0 ± 0.2 stones). A reduction in uric acid urinary excretion was noted as well [71]. It was also shown, that under in vitro conditions, *E. repens* had antiadhesive activity towards uropathogenic *Escherichia coli.* It inhibited the binding of *E. coli* to the surface of bladder cells (cell line T24). Decreased bacterial adhesion (IC_25_ 630 mg/mL) was observed, and this effect has been linked to the prevention of urinary tract infections [66]. In a clinical trial in patients with micturition disorders, an ethanolic extract of *Elymus repens* caused a reduction in urinary incontinence, nycturia, dysuria, and tenesmus due to adenoma of prostate, cystitis, and prostatitis [72].

### 2.13. Equisetum arvense *L.*

The genus *Equisetum* encompasses 30 species of rush-like, long-standing herbs. An extract from *Equisetum arvense* is used to relieve pain and increase diuresis. It is highly effective in removing water from the body. This effect is due to the action of various components, i.e., equisetonin, potassium, magnesium, calcium, ascorbic acid, and caffeic acid [100,101,102]. Recently, Carneiro et al. [35] noted that a standardized, dried *E. arvense* extract showed a diuretic effect stronger than hydrochlorothiazide in healthy male volunteers. In addition, *E. arvense* did not have a significant effect on the urinary excretion of catabolites and electrolytes. It was also deemed safe for oral consumption.

### 2.14. Foeniculum vulgare Mill.

*Foeniculum vulgare* is a biennial medicinal and aromatic plant. Its bulbs and fronds have a culinary application [103]. Bardai et al. [21] showed that administration of *F. vulgare* fruit extract significantly increased urinary volume, as well as potassium and sodium excretion. The dosage was chosen according to the dose used in Moroccan traditional medicine.

### 2.15. Fraxinus excelsior *L.*

The *Fraxinus* genus has been used in traditional medicine in many regions of the globe. It is valued for its diuretic properties and has been used in the treatment of rheumatic pain, arthritis, cystitis, constipation, dropsy, and itching scalp. In Polish folk medicine, the bark and leaves of *Fraxinus excelsior*, which is native to Europe, are used to treat various conditions, including wounds, diarrhea, and dysentery [42]. *F. excelsior* leaf extracts have been used to promote renal excretion. Aqueous and ethanolic extracts can be used to make spray-dried powders, which increase the excretion of chloride and sodium ions, potassium, and urea. The presence of flavonoids is likely the reason for their diuretic activity [43]. Moreover, the daily oral administration of aqueous extract from *Fraxinus excelsior* increased urination and promoted urinary excretion of sodium, potassium, and chlorides in hypertensive rats [44].

### 2.16. Humulus lupulus *L.*

*Humulus lupulus* is well-known across the world for its use in the brewing industry. The major compounds isolated from mature hop cones include chalcones, bitter acids, and terpenes [104]. Moreover, hops have a long history of use in folk medicine, particularly to treat sleep disturbances. In addition, some in vitro and in vivo experiments support the value of hops as a traditional antibacterial and antifungal remedy, while others show them to have diuretic activity.

### 2.17. Juniperus communis *L.*

In both folk and official medicine, juniper berries (*Juniperus communis*) are believed to have diuretic, antiseptic, stomachic, and carminative activities [11]. Juniper berries contain essential oil (between 0.5 and 2%), which is a source of phenolic compounds, carbohydrates, fatty acids, and sterols. However, Stanić et at. [73] found that the diuretic activity of juniper berries is the result of the action of essential oil and hydrophilic constituents.

### 2.18. Lycopodium clavatum *L.*

*Lycopodium clavatum* is a pteridophyte that grows abundantly in subtropical and tropical regions and in numerous European countries. This plant is used to help with digestion, relieve gastric inflammation, and for the treatment of chronic kidney disorders [105]. It has been shown that *L. clavatum* can play a role in alternative treatment of gout by inhibiting xanthine oxidase. Alcoholic and aqueous extracts (50–100 μg/mL) of the whole plant inhibited oxidase activity by 13–58% [106].

### 2.19. Nigella sativa *L.*

The seeds of *Nigella sativa* have been used in the Indian subcontinent, Europe, and Arabian countries for culinary purposes and as a remedy for several illnesses and conditions such as hypertension, asthma, inflammation, diabetes, bronchitis, cough, headache, fever, influenza, dizziness, and eczema. The seeds and seed oil are used as a carminative and in food as a spice [51], although they are seldom cultivated as a spice in Poland (atlas-roslin.pl). Zaoui et al. [50] showed, that dichloromethane extract from *N. sativa* seeds has a strong diuretic effect in rats with spontaneous hypertension. The diuretic activity of the extract was approximately 16% stronger than that of frusemide, which increased diuresis by 30%. Apart from increased diuresis, the excretion of sodium, chloride, potassium, and urea was higher as well.

### 2.20. Petroselinum crispum (Mill.) Fuss

*Petroselinum crispum*, commonly known as parsley, is used as carminative, gastro tonic, diuretic, and antiseptic of the urinary tract. In addition, in Bulgarian phytotherapy, various parts of *P. hortense* (leaves, seeds, and roots) are thought to have diuretic activity [107].

De Ribeiro et al. [74] investigated the action of aqueous-ethanol extract from the seeds of *P. sativum* which was administered to rats. Increased urinary volume and sodium excretion were observed, and these effects were similar to those of furosemide. Yarnell et al. [10] noted that *P. crispum* has strong potency which was compared to properties of other herbal diuretics, including *Taraxacum officinale* and *Ononis compestris*, whose potential was described as medium. The main active compounds of *P. sativum* are phenolics, which include flavonoids such as appinin and apigenin. These compounds possess a wide range of biological effects, diuretic activity among them [107]. Moreover, Kreydiyyeh and Usta [75] observed that aqueous parsley seed extract increased diuresis. Its mechanism of action appears to be mediated by the inhibition of Na^+^-K^+^ pump that leads to a reduction in Na^+^ and K^+^ reabsorption, which in turn, causes osmotic water flow into the lumen and diuresis. In another study, Alyami and Rabah [76] reported that parsley leaf tea induced no significant differences in urine volume, pH, sodium, chloride, magnesium, potassium, creatinine, urea, citric acid, or uric acid. It did not induce a significant reduction in urinary tract stone formation as well. Moreover, Saeidi et al. [77] showed that aqueous extracts of *P. sativum* had a therapeutic effect on calcium oxalate stones in rats and reduced the number of calcium oxalate deposits. It has been observed that *P. sativum* extract significantly increases the calcium level and decreases the magnesium level in serum.

### 2.21. Plantago major *L.*

*Plantago major*, a perennial herb, is found wild throughout the whole of Europe and temperate regions of Asia. Every part of the plant has been used in traditional medicines that are used to treat cough, diarrhea, dysentery, infections, pain, or urinary tract calculus [47]. An ethanolic extract of *P. major* significantly inhibited the growth of calcium oxalate crystals (dihydrate variety) in vitro. It has been shown, that *P. major* extract was better than allopurinol and potassium citrate in the reduction in the risk of unfavorable renal outcomes [78].

### 2.22. Rosa canina *L.*

*Rosa canina*, a plant native to Poland can also be found in other European countries [108]. Its fruit is extensively used worldwide in foods such as jelly and jam, in various beverages such as tea, and in traditional medicine to treat urate metabolism disorders. Urate is the end product of purine metabolism. It is produced from hypoxanthine after double enzyme catalysis by xanthine oxidase (XO), which is carried out in the liver. An imbalance in serum urate production and excretion induces hyperuricemia, which can also develop into gout and kidney stones, and accelerate the progression of renal diseases. In vitro studies have shown that *R. canina* extracts inhibited XO activity and significantly decreased the levels of serum urate eight hours after administration. It is suggested that *R. canina* hot water extract can serve as a functional food which could be beneficial for patients with a high level of urate. It could also be used in the treatment of hyperuricemic patients [109]. It has also been shown, that aqueous extract lowered the levels of renal and urinary calcium, decreased the number and size of kidney CaOx calculi, and promoted citrate excretion without affecting the urinary concentrations of oxalate, or urine volume and pH [79].

### 2.23. Sambucus ebulus *L.* and Sambucus nigra *L.*

*Sambucus ebulus* is a herbaceous plant well-known in traditional European medicine for its healing effects in many disorders; however, its toxicity limits its value as food [110]. Dimkov [16] indicates that a decoction from *S. ebulus* has diuretic and diaphoretic properties. Also, Beaux et al. [80] report increased excretion of sodium in rats after the administration of *S. nigra* flower extract. In addition, Walz and Chrubasik [81] indicate that *S. nigra* concentrate can be administered without the risk of adverse effects, even in patients afflicted with idiopathic nephrolithiasis. They observed that *S. nigra* concentrate did not change the urinary pH or hydrogen ion levels. The solubility of stone-inducing ions remained unaffected as well.

### 2.24. Taraxacum campylodes G.E.Haglund

*Taraxacum campylodes* is a widespread perennial of the Asteraceae family. It is commonly seen as a weed but contains a large variety of chemical compounds with healing potential. Most of the active substances found in *T. campylodes* are phenols and terpenes; however, carbohydrates, proteins, fatty acids, vitamins, and minerals are also present. This range of compounds has resulted in the plant being used as a natural drug in the treatment of gout and diarrhea, as well as problems associated with the bladder, spleen, and liver [111]. Leaf ethanol extract (1 g/mL) demonstrated a diuretic effect in a group of women treated with the extract every five hours for four days, with no side effects observed [112,113,114].

### 2.25. Urtica dioica *L.* and Urtica urens *L.*

*Urtica dioica* is a perennial plant widely distributed throughout the temperate and tropical regions of the globe, common throughout Poland [86]. Traditionally, *U. dioica* leaves and roots are known for a wide range of ethnomedicinal uses. *Urtica urens* has similar pharmacological properties [115]. The aqueous extract of the aerial part of *U. dioica* at a low dose (4 mg/kg/h) increased diuresis by 11%, and by 84% at a high dose (24 mg/kg/h). Moreover, low and high doses induced natriuresis by 28% and 143%, respectively [59]. Various preparations from *U. dioica* have been investigated experimentally, but only stinging nettle juice, tea, stew, encapsulated fresh freeze-dried leaf powder, and proprietary extracts have been used in human studies [115,116]. In vitro studies have shown that, like *Betula* spp. and *Elymus repens*, *Urtica* spp. shows an antiadhesive effect against the binding of uropathogenic *E. coli* to the surface of bladder cells (IC_25_ 630 mg/mL) [66]. However, while *U. dioica* ethanolic extract at a dose of 1g/kg (p.o) had no effect on diuresis, its administration at a dose of 500 mg/kg (i.p) resulted in a significant increase of the urine output [115].

### 2.26. Vaccinium myrtillus *L.*

*Vaccinium myrtillus* is a small deciduous shrub that is very popular in Poland and other European countries. The leaves are used in folk medicine as decoctions and infusions for treating conditions associated with the urinary tract thanks to their astringent and antiseptic properties. *Vaccinium vitis idaea* is another evergreen small shrub growing in Europe; its berries are known to have the same properties as *V. myrtillus* fruits, while the leaves have diuretic and urinary antiseptic activities, which have been attributed to their high concentration of tannins, arbutin, and arbutin derivatives [117].

### 2.27. Viola tricolor *L.*

*Viola tricolor* has a long history in phytomedicine. Its aerial parts have been described and used in Europe for centuries; they were utilized in the therapy of skin disorders and upper-respiratory problems and were used as a diuretic [60]. *V. tricolor* inhabits the lowland and lower mountain areas of Poland [118]. Its anti-inflammatory and diuretic properties have been attributed to the presence of saponins (5.98%) and flavonoids (1.81–1.99%). In a study conducted on rats, *Viola tricolor* tincture was found to have a moderate diuretic effect (diuretic index was 1.103, saluretic index of Na^+^ was 1.181, and saluretic index of K^+^ was 1.365) [61].

Figure 1 summarizes the activity of each plant listed in Section 2, while Figure 2 shows their mechanisms of action in urinary diseases and disorders.

## 3. Phytochemicals Important in the Treatment of Urinary Diseases

*Apium graveolens* L., *Arctostaphylos uva-ursi* (L.) Spreng, *Betula* spp., *Elymus repens* (L.) Gould, *Equisetum arvense* L., *Juniperus communis* L., *Levisticum officinale* W.D.J. Koch, *Ononis spinosa* L., *Petroselinum crispum* (Mill.) Fuss, *Solidago virgaurea* L., *Taraxacum campylodes* G.E.Haglund, *Urtica* spp., *Vaccinium myrtillus* L., *Vaccinium vitis-idaea* L., and *Viola tricolor* L. are all known to have diuretic and urinary tract disinfectant activity [18]. These herbs usually contain monosaccharides, flavonoids, volatile oils, saponins, terpenes, or tannins, which increase urine volume by promoting kidney blood flow and raising the glomerular filtration rate. Nevertheless, unlike synthetic diuretics, they do not reduce the resorption of Na^+^ and Cl^−^ in the renal tubules [11]. Lien et al. [122] identified the most common chemical ingredients in plants used in the treatment of kidney disease and/or kidney protection, and their possible mechanisms of action in traditional Chinese medicine. Antioxidant polyphenols can prevent nephropathy by interacting with free radicals or reactive oxygen species. Many antioxidants contain a component part bound to oxidizable functional groups like ferulic acid and isoferulic acid, tannins, flavonoids, and isoflavonoids [123].

Some essential fatty acids can exhibit both anti-inflammatory and pro-inflammatory properties, modulating the immune response [122]. Arbutin is a phenolic glycoside that shows antimicrobial and anti-inflammatory activity. It can decrease the production of inflammatory cytokines, and reduce the expression of inducible NO synthase (iNOS). About 65% of arbutin undergoes hydrolyzation to hydroquinone, which takes place primarily in the intestines. The main mechanism of the antimicrobial activity of hydroquinone is tied to the destruction of the bacterial cell wall [124]. The antimicrobial effect of quinones is partially attributed to their ability to form irreversible complexes with proteins, specifically with nucleophilic amino acids. They are thought to affect adhesins exposed on the surface, enzymes bound to membranes, and cell wall polypeptides [125,126].

Flavonoids are produced by plants partly in response to microbial infection. The mechanism of action of both flavonoids and tannins is thought to be similar to quinones. They inactivate microbial adhesions, cell envelope transport proteins, and enzymes, and possibly inactivate the microorganisms directly. It has been observed that tannins elicit antimicrobial activity against filamentous fungi, yeasts, and bacteria [125,126].

Rutin is a flavonoid that can be found in several plant species used in the treatment of urinary diseases, including *Fraxinus excelsior*, *Matricaria chamomilla*, or *Viola tricolor* [42,60,125]. It can decrease the levels of oxalate and calcium in the kidneys and urine. Oxalate and calcium are the main components of urinary stones. This effect is thought to be due to the inhibition of oxalate synthesis and the increase in calcium sequestration by nitric oxide [127,128]. Moreover, Kappel et al. [129] have shown, that rutin increases the uptake of calcium into skeletal muscles, which is mediated through mitogen-activated kinase (MEK) and protein kinase A (PKA) signaling pathways.

There are many different mechanisms of action of diuretic or aquaretic drugs. Based on their site of action, diuretics can be classified into thiazide (inhibition of the Na^+^/Cl^−^ transporter), loop (inhibition of the Na^+^/K^+^/2Cl^−^), potassium-sparing diuretics, and carbonic anhydrase inhibitors. Some aquaretics block the V2 vasopressin receptor reducing aquaporin 2 (AQP2) water channel and sodium/glucose transporter 2 (SGLT2) inhibitors. New mechanisms of action of diuretic compounds (for example, inhibition of adenosine A_1_ receptors or urea transporters) are also being discovered [130].

Luteolin (at a dose of 3 mg/kg) showed diuretic activity in normotensive and spontaneously hypertensive rats. Moreover, in normotensive rats, it did not change the calcium levels in urine, which is an important aspect of urinary stone formation prevention. Importantly, luteolin did not increase potassium excretion, unlike thiazide and loop diuretics. The authors suggest that its mechanism of action might be linked to muscarinic acetylcholine receptors [131]. Similar to luteolin, gallic acid also increased diuresis without increasing the potassium excretion in rats at a dose of 3 mg/kg. The effect was similar to that of hydrochlorothiazide (a clinical reference thiazide-type diuretic). However, here, the effect was not dependent on the activation of muscarinic acetylcholine receptors. Instead, the mechanism of action of gallic acid might be linked to endogenous prostanoid generation [132].

Triterpene saponins can stimulate microcirculation due to their surfactant properties. The diuretic action of these compounds is believed to be associated with local irritation of kidney epithelia. The diuresis caused by plants such as *Ononis* spp., *Betula* spp., and *Solidago* species is relatively mild, and the effect might originate from the accompanying flavonoids and essential oils. An alternative theory is that the potassium content of these plants is, in fact, the diuretic agent [133]. Unfortunately, the mechanisms of action of many plant-derived diuretic compounds remain unknown.

Table 3 summarizes the most commonly used plants in Polish folk medicine for the treatment of urinary diseases, and their main phytochemicals. Flavonoids were found in every plant species listed in the table. Phenolic acids, coumarins, tannins, and terpenes were also common.

The chemical composition of many plants used in folk medicine to treat urinary diseases remains unclear, as does the composition of plants used in contemporary medicine. Experimental data on the pharmacological effects of these plants are insufficient. Some of the plants from the traditional folk pharmacopeia are still in use, but a large group has been discarded, and the body of scientific evidence on the effectiveness and safety of their use can be sparse. A precise chemical analysis of the composition of these plants, based on in vivo and in vitro studies, may allow for a rediscovery of valuable therapeutics for the treatment of urinary diseases. The following plants have a long history of use in folk medicine; however, the literature indicates they are rarely used in contemporary Polish phytomedicine and merit further analysis: *Allium ursinum* L., *Angelica sylvestris* L., *Bryonia alba* L., *Elsholtzia ciliata* (Thunb.) Hyl., *Galium aparine* L., *Onopordum acanthium* L., *Quercus robur* L., *Raphanus raphanistrum* subsp. *sativus* (L.) Domin, *Sanguisorba officinalis* L., *Silene vulgaris* (Moench) Garcke, *Stellaria media* (L.) Vill., *Trifolium arvense* L., and *Trigonella caerulea* (L.) Ser. These plants have been used in folk medicine, but according to literature sources, are not used in contemporary Polish phytomedicine [134].

The main chemical compounds of selected plants used in Polish folk medicine to treat urinary diseases and their potential molecular targets of action are presented in Figure 3.

## 4. Conclusions

There are many plants used to treat urological diseases in Polish folk medicine. They are prescribed for various conditions, including UTIs, urolithiasis, ischuria, edema, or cystitis. Currently, both in vitro and in vivo studies have confirmed that many of these plant species have beneficial properties, such as diuretic, antihyperuricemic, antimicrobial, and anti-inflammatory activity, or prevention of urinary stone formation. These effects are exerted through different mechanisms, for example, through the activation of bradykinin B2 receptors, the inhibition of xanthine oxidase, or the inhibition of Na^+^-K^+^ pumps. Many plants used in folk medicine are rich in phytochemicals with proven effectiveness against urinary tract diseases, such as rutin, arbutin, or triterpene saponins. The present study constitutes a good basis for future comparison of Polish folk and contemporary medicine, with the aim of restoring the use of phytochemicals with proven activities. Many of the plants presented above (e.g., *Petroselinum crispum* (parsley), *Ribes nigrum* (blackcurrant), and *Apium graveolens* (celery root)) are common components of the everyday diet, which makes them easily accessible for people who are ill or at risk of urinary tract diseases. Furthermore, plant-based medicines which have been used for generations can serve as a basis for creating new, inexpensive, and safe drugs. Conducting well-controlled and high-quality human clinical experiments in this area is encouraged.

## Figures and Tables

**Figure 1 pharmaceuticals-17-00435-f001:**
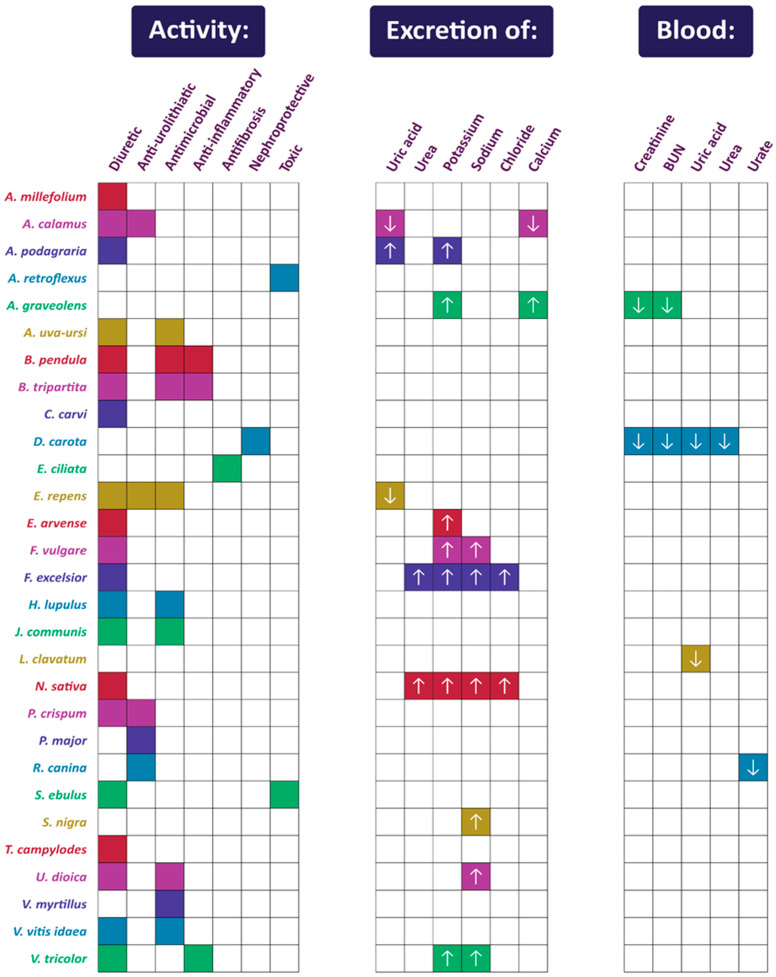
Summary of activities of plants used in Polish folk medicine to treat urinary disorders (confirmed by scientific studies). The arrows indicate an increase (upward-facing arrow) or decrease (downward-facing arrow) in the excretion or blood concentration of compounds and markers. Compilation of data from Section 2.

**Figure 2 pharmaceuticals-17-00435-f002:**
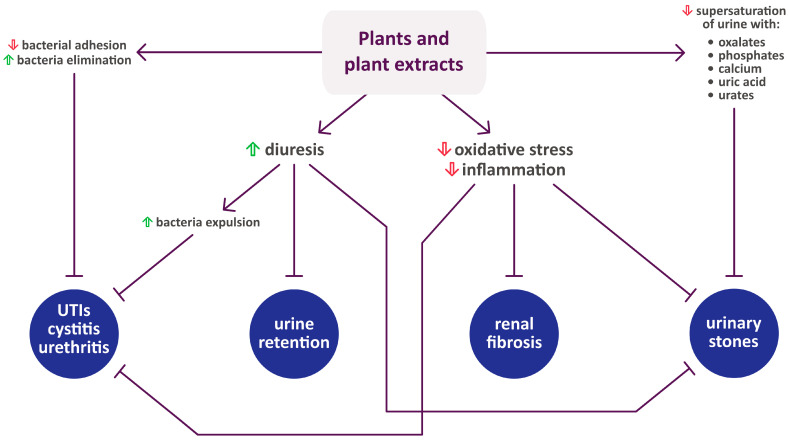
The mechanisms of action of plants toward urinary diseases and disorders. Upward-facing green arrow indicates increase, while downward-facing red arrow indicates decrease. Compilation of data from Section 1 and Section 2, as well as [119,120,121]. UTIs—urinary diseases.

**Figure 3 pharmaceuticals-17-00435-f003:**
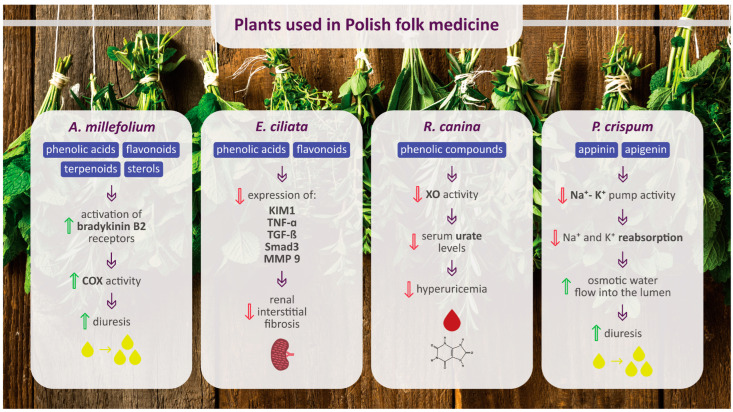
Main chemical compounds of selected plants used in Polish folk medicine to treat urinary diseases and their potential molecular targets of action. Upward-facing green arrows indicate an increase, while downward-facing red arrows indicate a decrease. COX, cyclooxygenase; KIM1, kidney injury molecule 1; TNF-α, tumor necrosis factor alpha; TGF-β, transforming growth factor β; MMP 9, matrix metalloproteinase 9; XO, xanthine oxidase.

**Table 1 pharmaceuticals-17-00435-t001:** Ethnopharmacological characteristics of plants reported for the treatment of urinary diseases in Poland.

Name of the Species (Family)	Polish Name	English Common Name	Plant Part	Form of Administration in Folk Medicine	Utilization	References
*Acorus calamus* L. (Acoraceae)	tatarak pospolity	sweet flag	Rz	infusion, decoction	diuretic (urinary stones)	[15]
*Sambucus ebulus* L. (Adoxaceae)	bez hebd	dwarf elder	Ro	infusion, decoction	diuretic (kidney diseases and edema)	[16]
*Sambucus nigra* L. (Adoxaceae)	dziki bez czarny	elder	Fl, Ba, Lf	infusion, as tea	diuretic (urethritis)	[17]
*Apium graveolens* L. (Apiaceae)	seler zwyczajny	celery root	Lf, Ro	infusion, as tea	diuretic (bladder diseases)	[18]
*Carum carvi* L. (Apiaceae)	kminek zwyczajny	meridian fennel	Fr	decoction, as tea	diuretic	[19]
*Daucus carota* L. (Apiaceae)	marchew zwyczajna	Queen Anne’s lace	Ro	fresh	diuretic (edema)	[20]
*Foeniculum vulgare* Mill. (Apiaceae)	koper włoski	fennel	Fr	fresh, decoction, as tea	diuretic	[21]
*Levisticum officinale* W.D.J. Koch (Apiaceae)	lubczyk ogrodowy	lovage	Ba, Lf	decoction	diuretic (cystitis)	[18]
*Petroselinum crispum* (Mill.) Fuss (Apiaceae)	pietruszka zwyczajna	parsley	WP	fresh, infusion, decoction	diuretic (ischuria)	[18]
*Hedera helix* L. (Araliaceae)	bluszcz pospolity	ivy	Lf	infusion	anti-inflammatory (kidney stones)	[22]
*Asarum europaeum* L. (Aristolochiaceae)	kopytnik pospolity	asarabacca	Rz	decoction	analgesic (cystitis, edema)	[23]
*Betula pendula* Roth (Betulaceae)	brzoza brodawkowata	silver birch	Lf	infusion, as tea	diuretic (cystitis)	[18]
*Betula pubescens* Ehrh. (Betulaceae)	brzoza omszona	downy birch	Lf	infusion	diuretic (cystitis)	[18]
*Raphanus raphanistrum* subsp. *sativus* (L.) Domin (Brassicaceae)	rzodkiew świrzepa	wild radish	Ro	fresh	diuretic (bladder diseases)	[24]
*Humulus lupulus* L. (Cannabaceae)	chmiel zwyczajny	hop	Fl	decoction	diuretic (cystitis)	[25]
*Achillea millefolium* L. (Compositae)	krwawnik pospolity	yarrow	AP	infusion	analgesic (bladder and kidney diseases)	[26]
*Bidens tripartita* L. (Compositae)	uczep trójlistkowy	three-lobe beggarticks	AP	infusion	diuretic (ischuria, urinary stones)	[27,28]
*Carlina acaulis* L. (Compositae)	dziewięćsił bezłodygowy	stemless carline thistle	Ro	decoction	diuretic, anti-inflammatory (ischuria, urethritis, cystitis)	[29]
*Cichorium intybus* L. (Compositae)	cykoria podróżnik	chicory	Ro	decoction	diuretic (ischuria)	[30]
*Matricaria chamomilla* L. (Compositae)	rumianek pospolity	chamomile	In	infusion, as tea	anti-inflammatory (nephritis, urethritis, cystitis)	[31]
*Onopordum acanthium* L. (Compositae)	popłoch pospolity	cotton thistle	Lf	infusion	anti-inflammatory (inflammation of the urinary tract)	[32,33]
*Solidago virgaurea* L. (Compositae)	nawłoć pospolita	European goldenrod	FS	decoction	diuretic (kidney stones, cystitis)	[18]
*Taraxacum campylodes* G.E.Haglund (Compositae)	mniszek lekarski	dandelion	Ro	decoction	diuretic (edema)	[18]
*Bryonia alba* L. (Cucurbitaceae)	przestęp biały	white bryony	Ro	decoction	diuretic (edema)	[34]
*Juniperus communis* L. (Cupressaceae)	jałowiec pospolity	juniper	JB	infusion, decoction	diuretic (urinary diseases, hematuria, edema)	[18]
*Equisetum arvense* L. (Equisetaceae)	skrzyp polny	horsetail	AP	decoction	diuretic (kidney stones, urine retention)	[18,35]
*Arctostaphylos uva-ursi* (L.) Spreng. (Ericaceae)	mącznica lekarska	bearberry	Lf	decoction	antiseptic (kidney stones, inflammation of the urinary tract)	[36]
*Vaccinium myrtillus* L. (Ericaceae)	borówka czarna	bilberry	Lf, Fr	decoction, fresh (fruit)	diuretic (urine retention)	[18]
*Vaccinium vitis-idaea* L. (Ericaceae)	borówka brusznica	lingonberry	Lf, Fr	decoction, fruit juice	diuretic, antiseptic (kidney stones, inflammation of the urinary tract)	[18]
*Quercus robur* L. (Fagaceae)	dąb szypułkowy	pedunculate oak	Ba	decoction	anti-inflammatory (urethritis, cystitis)	[37]
*Ribes nigrum* L. (Grossulariaceae)	porzeczka czarna	blackcurrant	AP, Fr	infusion, juice, as tea	anti-inflammatory (kidney stones, cystitis, kidney diseases)	[38]
*Hypericum perforatum* L. (Hypericaceae)	dziurawiec zwyczajny	perforate St John’s-wort	AP	infusion, decoction, as tea	diuretic (kidney diseases)	[39]
*Lamium album* L. (Lamiaceae)	jasnota biała	white nettle	Fl	infusion	anti-inflammatory (urethritis, cystitis)	[40]
*Cytisus scoparius* (L.) Link (Leguminosae)	żarnowiec miotlasty	broom	AP	decoction	diuretic (edema)	[41]
*Ononis spinosa* L. (Leguminosae)	wilżyna ciernista	spiny restharrow	Ro	decoction	diuretic (kidney/bladder stones)	[18]
*Fraxinus excelsior* L. (Oleaceae)	jesion wyniosły	ash	Ba, Lf	decoction, infusion	diuretic, anti-inflammatory (urethritis, cystitis)	[42,43,44]
*Fumaria officinalis* L. (Papaveraceae)	dymnica pospolita	fumitory	AP	infusion	diuretic (bladder diseases)	[45]
*Plantago lanceolata* L. (Plantaginaceae)	babka lancetowata	English plantain	AP	infusion	anti-inflammatory (cystitis)	[46]
*Plantago major* L. (Plantaginaceae)	babka zwyczajna	broadleaf plantain	AP, Lf	infusion	diuretic (urological diseases)	[47]
*Elymus repens* (L.) Gould (Poaceae)	perz właściwy	couch grass	Rz	decoction, infusion	diuretic (edema, urinary stones)	[18]
*Polygonum bistorta* L. (Polygonaceae)	rdest wężownik	bistort	AP, Rz	infusion	anti-inflammatory (urethritis, cystitis, hematuria)	[48]
*Polypodium vulgare* L. (Polypodiaceae)	paprotka zwyczajna	polypody	Rz	infusion	diuretic (kidney diseases)	[49]
*Nigella sativa* L. (Ranunculaceae)	czarnuszka siewna	black caraway	Se	infusion	diuretic, anti-inflammatory (urethritis, cystitis)	[50,51]
*Filipendula ulmaria* (L.) Maxim. (Rosaceae)	wiązówka błotna	meadowsweet	Fl	infusion	diuretic, anti-inflammatory (urethritis, cystitis)	[52]
*Fragaria vesca* L. (Rosaceae)	poziomka pospolita	wild strawberry	AP, Fr	decoction, as tea	diuretic (kidney/bladder stones)	[53]
*Potentilla anserina* L. (Rosaceae)	pięciornik gęsi	silverweed	AP	infusion	anti-inflammatory, diuretic (edema)	[54]
*Prunus avium* (L.) L. (Rosaceae)	wiśnia ptasia	wild cherry	LS	infusion	diuretic (urinary stones)	[55]
*Sorbus aucuparia* L. (Rosaceae)	jarząb pospolity	rowan	Fr, Fl	infusion, jam	diuretic (urinary stones)	[56]
*Galium aparine* L. (Rubiaceae)	przytulia czepna	cleavers	AP	infusion	diuretic (kidney stones)	[57]
*Ruta graveolens* L. (Rutaceae)	ruta zwyczajna	herb-of-grace	Lf	infusion	diuretic (edema)	[58]
*Urtica dioica* L. (Urticaceae)	pokrzywa zwyczajna	stinging nettle	WP	infusion, as tea	diuretic (urinary stones)	[59]
*Viola tricolor* L. (Violaceae)	fiołek trójbarwny	heartsease	AP	infusion, as tea	diuretic (cystitis)	[60,61]

Plant parts: AP, aerial parts; Ba, bark; Fl, flower; FS, flowering shoots; Fr, fruit; In, inflorescence; JB, juniper berry; LS, leaf stalk; Lf, leaf; Rz, rhizome; Ro, root; Se, seed; Sp, spores; WP, whole plant.

**Table 2 pharmaceuticals-17-00435-t002:** Summary of research reporting effects on urinary volume and sodium excretion. The plants are listed alphabetically.

Scientific Name	Model	Extract Used	Dose	Control/ Reference	Duration/ Total Number of Animals or Participants	UV	UNa	Other Effects Found	Citations
*Achillea millefolium* L.	conscious rats—Wistar	aqueous	125, 250, 500 mg/kg	5% Tween 80 aqueous solution (1 mL/kg) /HCTZ (10 mg/kg)	8 h/50 (5 per group)	0	0	-	[62]
		hydro-ethanolic	30–300 mg/kg			+ve	+ve		
		dichloromethane subfractions	10, 30 mg/kg			+ve	+ve		
*Acorus* *calamus* L.	conscious rats—Wistar	ethanolic	250, 500, 750 mg/kg	sodium carboxymethyl cellulose (0.5%; 10 mL/kg)/furosemide (15 mg/kg)	7 h/30 (6 per group)	+ve	+ve	-	[15]
	urolithiatic rats	ethanolic	750 mg/kg	sodium carboxymethyl cellulose (1%)/Cystone (750 mg/kg, p.o.)	28 d/24 (6 per group)	+ve	−ve	antiurolithiatic	
*Aegopodium* *podagraria* L.	conscious mice	essential oil	1 mg/kg	the initial state/ olimetin (1 mg/kg)	3 d	+ve	-	uricosuric	[63]
	conscious rats	aqueous	1g/kg	water/HCTZ (20 mg/kg)	8 w/42 (6 per group)	−ve	+ve	-	[64]
		tincture	1 and 5 mL/kg			0	+ve		
*Apium graveolens* L.	conscious rabbits	fresh celery	8 g/kg	water/-	10 d/16 (8 per group)	+ve	+ve	-	[65]
*Betula* spp.	in vitro bladder cell line T24	hydro-ethanolic	9.4%	-	no data	-	-	antiadhesive against UPEC strain 2980	[66]
*Bidens* *tripartita* L.	Human	aqueous	1 mL/kg	-	2 m/24	+ve	-	-	[67]
*Carum carvi* L.	conscious rats—Wistar	aqueous	100 mg/kg	water/ furosemide (10 mg/kg)	1–24 h/20 (5 per group)	+ve	+ve	-	[68]
					8d	+ve	+ve		
*Daucus carota* L.	conscious rats—Wistar	crude extract	200, 400 mg/kg/day	normal saline (i.p.) and 0.5% carboxymethyl cellulose/-	8 d/24 (6 per group)	-	-	nephro- protective	[69]
*Elsholtzia ciliata* (Thunb.) Hyl.	conscious rats— Sprague Dawley	ethanolic	300 and 500 mg/kg	water/ captopril 200 mg/kg	14 d/50 (10 per group)	-	-	anti-inflammatory, protective against renal fibrotic disease	[70]
*Elymus* *repens* (L.) Gould	human	dry extract/twice a day	100 mg	combination of different drugs	5 m/50 (25 per group)	-	0	protective against renal stone formation	[71]
	in vitro bladder cell line T24	hydro-ethanolic	9.5%	-	-	-	-	antiadhesive against UPEC strain 2980	[66]
	human	ethanolic	20% (60 drops 3 times daily)	-	99	-	-	mictirition problems reduction	[72]
*Equisetum arvense* L.	human	dry extract	900 mg/day	corn starch, 900 mg/day or HCTZ (25 mg/day)	4 d each stage/10 d washout interval/ 36 (6 per group)	+ve	-ve	-	[35]
*Foeniculum vulgare* Mill.	conscious rats—SHR and Wistar-Kyoto	dry	190 mg/kg	water	5 d/61	+ve	+ve (in SHR)	-	[21]
*Fraxinus excelsior* L.	conscious rats—Wistar	spray-dried powders from aqueous and alcoholic extracts	no data	water + 3% gum arabic	6 h/25 (5 per group)	-	+ve	-	[43]
	conscious rats—SHR and Wistar-Kyoto	aqueous	20 mg/kg	water	21 d/25 (5 per group)	+ve	+ve	-	[44]
*Juniperus communis* L.	conscious rats—Wistar	0.1% water solution of essential oil and 0.01% water solution of terinen-4-ol	5 mL/100 g b.w.	water or water + 0.2% Tween 20/Moduretic (5 mg HCTZ and 50 mg amyloride)/ ADH	3 d/28 (min. 5 per group)	+ve	-	-	[73]
*Nigella* *sativa* L.	conscious rats—SHR	dichloromethane extract	0.6 mL/kg/day	furosemide (5 mg/kg/day)	15 d	+ve	+ve	-	[50]
*Petroseli-num* *crispum* (Mill.) Fuss	conscious rats—Wistar	Hydro-ethanolic (50:50, *v*/*v*)	40 mL/kg	0.9% NaCl, 40 mL/kg	4 h	+ve	+ve	-	[74]
	conscious rats— Sprague–Dawley	aqueous 20%	-	water/ furosemide (0.6 mM) and amiloride (1mM)	24 h/6	+ve	-	inhibition of Na^+^-K^+^ pump	[75]
	human	aqueous (tea)	1200 mL/day	bottled water	2 w/20 (10 per group)	0	0	-	[76]
	conscious rats—Wistar	aqueous	200 and 600 mg/kg	untreated/1% ethylene glycol	30 d/36 (6 per group)	-	-	antiurolithiatic	[77]
*Plantago major* L.	in vitro calcium oxalate crystals	ethanolic	100 ppm to 350 ppm	water + DMSO/ allopurinol and potassium citrate	-	-	-	-	[78]
*Rosa canina* L.	conscious rats—Wistar	aqueous	65 mg/ mL	water/ potassium citrate	30 d/50 (10 per group)	-	-	antinephrolithiatic	[79]
*Sambucus nigra* L.	conscious rats— Sprague–Dawley	aqueous	50 mg/kg	0.45% saline/HCTZ (10 mg/kg)	24 h/5 groups	+ve	+ve	-	[80]
	human	Concentrate (120 g berries + flower juice and extract from 3.9 g of dried flowers)	200 mL/ day	-	7 d/11	-	-	lack of effect on urine pH	[81]
*Taraxacum campylodes* G.E.Haglund	conscious rats and mice	aqueous	0.5–6% (50 mL/kg)	No data/furosemide (80 mg/kg)	30 d/40 (20 per group)	+ve	+ve	-	[82]
*Urtica* *dioica* L.	anaesthetized rats—Wistar	aqueous hydro-ethanolic	4 mg/kg/h or 24 mg/kg/h	0.9% NaCl/ furosemide (2 mg/kg/h)	30 min/22	+ve	+ve	-	[59]
	in vitro bladder cell line T24		11.2%	-	-	-	-	antiadhesive against UPEC strain 2980	[66]

ADH, antidiuretic hormone; b.w., body weight; d, day; h, hour; HCTZ, hydrochlorothiazide; m, month; SHR, spontaneously hypertensive rats; UNa, urinary sodium; UPEC, uropathogenic *Escherichia coli*; UV, urine volume; w, week; +ve and −ve, significant increases and decreases, respectively; 0, no change; -, not measured.

**Table 3 pharmaceuticals-17-00435-t003:** The most commonly used plants in Polish folk medicine for the treatment of urinary diseases, and their main phytochemicals.

Specific Name	The Most Important Phytochemicals	References
*Apium graveolens* L.	flavonoids (apiin, apigenin, isoquercitrin), coumarins (apiumetin, apigravin, apiumoside, bergapten, celereoside, celerin), volatile oils (limonene, selenine), choline ascorbate, fatty acids	[10,125]
*Arctostaphylos uva-ursi* (L.) Spreng.	flavonols (myricetin, quercetin), iridoids (asperuloside, monotropein), phenolic glycosides (arbutin, methyl-arbutin), tannins (corilagin pyranoside, ellagic and gallic acids), terpenoids (amyrin, lupeol, uvaol, ursolic acid)	[10,125]
*Betula pendula* Roth	flavonoids (luteolin, myricetin, quercetin), saponins, tannins, triterpenes, volatile oil (α-betulenol), resin, chlorogenic acid	[10,18]
*Elymus repens* (L.) Gould	carbohydrates (glucose, fructose, mannitol, inositol, mucilaginous substances, triticin, pectin), flavonoids (tricin), volatile oils (agropyrene)	[125]
*Equisetum arvense* L.	flavonoids (luteolin, quercetin, apigenin, kaempferol), acids (coffee-tartaric acid, chlorogenic acid), triterpenoids (taraxerol isobauerenol, germanicol, oleanolic acid, ursolic acid, and betulinic acid), saponins, alkaloids (nicotine, palustrine, palustrinine), silicon, potassium	[18,102,122]
*Fraxinus excelsior* L.	flavonoids (quercetin, rhamnetin, rutin, isoquercetrin, kaempferol, astragalin), coumarin glucosides (esculin, fraxin), iridoids, coumarins, lignans, phenolic acids (*p*-hydroxybenzoic acid, protocatehuic acid, vanillic acid), phenylpropanoid glucosides (coniferin, syringin), triterpenes, mannitol	[42]
*Juniperus communis* L.	acids (ascorbic acid, diterpene acids, glucuronic acid), flavonoids (quercetin, isoquercitrin, amentoflavone, apigenin), tannins (gallocatechin, epigallocatechin, proanthocyanidins), volatile oils (pinene, myrcene, sabinene), junionone, resins	[125]
*Lycopodium clavatum* L.	flavonoids (apigenin), acids (vanillic, coumaric, ferulic acids and syringic acid), alkaloids (lycopodine)	[105,122]
*Matricaria chamomilla* L.	coumarins (umbelliferone, herniarin), flavonoids (apiin, apigenin, apigetrin, quercetin, quercimeritrin, luteolin, rutin), vilatile oils (α-bisabolol, chamazulene), amino acids, anthemic acid, choline, polysaccharide, tannin, triterpene hydrocarbons	[125]
*Ononis spinosa* L.	triterpenoids, flavonoids, tannins, volatile oil (carvone, trans-anethole)	[10,18]
*Petroselinum crispum* (Mill.) Fuss	flavonoids (glycosides of apigenin, luteolin), furanocoumarins (bergapten, oxypeucedanin), volatile oil (myristicin, apiole, tetramethoxyallylbenzene, terpenes, alcohols, aldehydes, ketones), proteins, fixed oil, oleo-resin, carbohydrates, vitamins (especially C and A)	[10,125]
*Plantago major* L.	acids (caffeic acid, benzoic acid, chlorogenic acid, *p*-coumaric acid, cinnamic acid, ferulic acid, fumaric acid, neochlorogenic acid, gentisic acid, *p*-hydroxybenzoic acid, syringic acid, salicylic acid, ursolic acid, oleanolic acid, vanillic acid, ascorbic acid), amino acids (asparaginę, DL-α-alanine, L-histidine, DL-leucine, DL-lysine, tryptophan, serine), alkaloids (boschniakine) carbohydrates, flavonoids (baicalein, apigenin, scutellarein, homoplantaginin, baicalin, nepitrin, hispidulin, luteolin, plantagoside), iridoids (aucubin), tannins, choline, allantoin, invertin, and emulsin	[125]
*Sambucus nigra* L.	flavonols (kaempferol, quercetin), triterpenes (α- and β-amyrin, ursolic and oleanolic acids), volatile oils (alkanes, fatty acids), tannin, chlorogenic acid, mucilage, pectin, sugar, and plastocynin	[125]
*Solidago virgaurea* L.	flavonoids (kaempferol, quercetin, isorhamnetin), saponins, acids (chlorogenic acid, caffeic acid, ferulic acid), tannins, volatile oils, fructans	[10,18]
*Taraxacum campylodes* G.E.Haglund	acids and phenols (*p*-hydroxyphenylacetic acid, caffeic acid, chlorogenic acid, monocaffeoyl tartaric acids, cichoric acid, taraxacoside, linolenic acid, linoleic acid, oleic acid, and palmitic acid), coumarins (aesculin, cichoriin), flavonoids (luteolin-7-diglucosides, luteolin-7-glucoside), resin, potassium, terpenoids (eudesmanolides, sesquiterpene lactones taraxinic acid esterified with glucose), carotenoids, vitamin A, choline, pectin, inulin, phytosterols	[10,125]
*Urtica dioica* L.	acids (caffeic, carbonic, caffeoylmalic, chlorogenic, silicic, formic, citric, glyceric, fumaric, malic, phosphoric, oxalic, quinic, succinic), amines (acetylcholine, betaine, choline, histamine, lecithin, serotonin), flavonol glycosides (isorhamnetin, kaempferol, quercetin), lignans, choline acetyltransferase, scopoletin, β-sitosterol, tannin	[10,125]
*Vaccinium myrtillus* L.	Fruit: anthocyanins (glycosides of delphinidin, petunidin, cyanidin, peonidin, malvidin), polyphenols (catechin, epicatechin, tannins), pectins, vitamin C	[125]
	Leaf: flavonoids (quercetin and its glycosides), phenolic acids (*p*-coumaric, caffeic, *p*-hydroxybenzoic), tannins, iridoids	
*Vaccinium vitis-idaea* L.	phenolic glycosides (arbutin), tannins, flavonoids (quercetin, myricetin), triterpenoids (ursolic acid)	[18,60]
*Viola tricolor* L.	flavonoids (violaquercitrin, rutin, violanthin, saponaretin, scoparin, orientin, vicenin, anthocyanidin glycosides), coumarins (umbelliferone), saponins, phenol carboxylic acids (protocatechuic acid, trans-caffeic acid, *p*-coumaric acid), salicylic acid and its derivatives such as the methyl ester and violutoside, mucilages, tannins, carotenoids (violaxanthin), ascorbic acid	[60]

## Data Availability

Data sharing is not applicable.

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
