# Peer review of "New Light on Plants and Their Chemical Compounds Used in Polish Folk Medicine to Treat Urinary Diseases"

_pharmaceuticals, 2024, doi:10.3390/ph17040435_

Round 1
Reviewer 1 Report
Comments and Suggestions for Authors
In this manuscript, authors have studied about “New light on plants and their chemical compounds used in Polish folk medicine to treat urinary diseases” and have reported the information about the plants used to treat urological diseases in Polish folk medicine. It is a good piece of research but the manuscript's quality does not meet Journal’s standards.
Overall, the manuscript in its current form is not acceptable for publication in the esteemed “Pharmaceuticals” journal and hence cannot be recommended for publication. The submitted manuscript has weak objective and hypothesis, flaws in organization of manuscript, failure to adhere to the theme, content quality, lack of interpretations and discussion are not enough significant for the publication in “Pharmaceuticals” journal.
Comments on the Quality of English Language
Quality of the English Language needs improvement.
Author Response
In this manuscript, authors have studied about “New light on plants and their chemical compounds used in Polish folk medicine to treat urinary diseases” and have reported the information about the plants used to treat urological diseases in Polish folk medicine. It is a good piece of research but the manuscript's quality does not meet Journal’s standards.
Overall, the manuscript in its current form is not acceptable for publication in the esteemed “Pharmaceuticals” journal and hence cannot be recommended for publication. The submitted manuscript has weak objective and hypothesis, flaws in organization of manuscript, failure to adhere to the theme, content quality, lack of interpretations and discussion are not enough significant for the publication in “Pharmaceuticals” journal.
Response: Thank you for reviewing the manuscript and for providing such helpful comments. All of them have been taken into consideration when revising the manuscript. For example,
We corrected the abstract.
We shortened the introduction.
We added new Figure to the manuscript (Figure 1) that shows the activity of each medicinal plant to summarize the information and make it more accessible.
We added more information about data collection: “The present paper collects information about plants used to treat urological diseases in Polish folk medicine, and reviews their pharmacological activities, most important phytochemicals, and mechanisms of action. We described 77 species belonging to 37 plant families. The greatest number of plant species used to treat urinary diseases were found to come from the Compositae (10 species), Rosaceae (9 species), Apiaceae (7 species), and Leguminosae (6 species). The plants were identified based on ethnopharmacological studies, Polish herbalist literature, and interviews with Polish traditional healers. We used the PubMed, Scopus, Google Scholar, Web of Science, Science Direct, and other databases to search for the ethnomedicinal uses of these plants, as well as phytochemistry, pharmacology, toxicity, and clinical studies. The search terms included the name of plant or phytochemical and the words “diuretic”, “urinary”, “UTI”, or “urolithiasis”. Plant names have been checked and updated with the online website The Plant List version 1.1 (www.theplantlist.org) of the Royal Botanic Gardens, Kew, and Missouri Botanical Garden, accessed between April and July 2023, and the online website Atlas of Vascular Plants of Poland (http://www.atlas-roslin.pl). The common names of the plants were identified by referring to research articles.”
We added more information about the phytochemicals responsible for the activities described in the manuscript and their mechanisms of action.
Reviewer 2 Report
Comments and Suggestions for Authors
Dear authors,
After reviewing the following manuscript entitled " New light on plants and their chemical compounds used in Polish folk medicine to treat urinary diseases” (Pharmaceuticals-2910361), I sent the following comments and observations that the authors should attend to before its publication in this journal.
First, authors must use the journal template. The list of abbreviations will be placed at the end of the manuscript.
The abstract is too long and without relevant information related to the content of the manuscript.
The introduction is too long. It should be summarized in a maximum of half a page.
The content of the second part must be written in the form of a table. The description is difficult to read and the information is much more accessible in a table.
I recommend that the pharmacological activity of each medicinal plant be clearly highlighted in the manuscript.
Since it is a review, I suggest making a prisma flow for documentation.
Authors must mention which data collection method they used. It should be mentioned from the databases used what is the situation of the articles that have botanical products under study.
References older than 10 years should be removed.
Author Response
After reviewing the following manuscript entitled " New light on plants and their chemical compounds used in Polish folk medicine to treat urinary diseases” (Pharmaceuticals-2910361), I sent the following comments and observations that the authors should attend to before its publication in this journal.
Response: Thank you for reviewing the manuscript and for providing such helpful comments. All of them have been taken into consideration when revising the manuscript.
First, authors must use the journal template. The list of abbreviations will be placed at the end of the manuscript.
Response: We corrected this. Now, the list of abbreviations is at the end of the manuscript.
The abstract is too long and without relevant information related to the content of the manuscript.
Response: We corrected the abstract.
The introduction is too long. It should be summarized in a maximum of half a page.
Response: We shortened the introduction.
The content of the second part must be written in the form of a table. The description is difficult to read and the information is much more accessible in a table.
Response: Table 2 summarizes the results of studies in section 2. However, the text also includes some additional information that we think would be impractical to include in a table. However, we added new Figure to the manuscript (Figure 1) that shows the activity of each medicinal plant to summarize the information and make it more accessible.
I recommend that the pharmacological activity of each medicinal plant be clearly highlighted in the manuscript.
Response: We added a new Figure (Figure 1) to clearly show pharmacological activity of each plant.
Since it is a review, I suggest making a prisma flow for documentation.
Response: Because this was not intended to be a systematic review, we did not collect data needed for prisma flow diagram. However, we added more information about our search method.
Authors must mention which data collection method they used. It should be mentioned from the databases used what is the situation of the articles that have botanical products under study.
Response: We added more information about data collection: “The present paper collects information about plants used to treat urological diseases in Polish folk medicine, and reviews their pharmacological activities, most important phytochemicals, and mechanisms of action. We described 77 species belonging to 37 plant families. The greatest number of plant species used to treat urinary diseases were found to come from the Compositae (10 species), Rosaceae (9 species), Apiaceae (7 species), and Leguminosae (6 species). The plants were identified based on ethnopharmacological studies, Polish herbalist literature, and interviews with Polish traditional healers. We used the PubMed, Scopus, Google Scholar, Web of Science, Science Direct, and other databases to search for the ethnomedicinal uses of these plants, as well as phytochemistry, pharmacology, toxicity, and clinical studies. The search terms included the name of plant or phytochemical and the words “diuretic”, “urinary”, “UTI”, or “urolithiasis”. Plant names have been checked and updated with the online website The Plant List version 1.1 (www.theplantlist.org) of the Royal Botanic Gardens, Kew, and Missouri Botanical Garden, accessed between April and July 2023, and the online website Atlas of Vascular Plants of Poland (http://www.atlas-roslin.pl). The common names of the plants were identified by referring to research articles.”
References older than 10 years should be removed.
Response: It is true that some references in this article are older than 10 years, but we feel that they are still relevant to this study, as they discuss the use of different plants in Polish folk medicine. To fix this issue, we have added some more recent references to support them – for example Szekalska et al., 2019, or Das, 2020.
Reviewer 3 Report
Comments and Suggestions for Authors
Reviewer's Comments on Manuscript-Pharmaceuticals-2910361-Peer-Review-V1
The manuscript entitled “New light on plants and their chemical compounds used in Polish folk medicine to treat urinary diseases” is review article based on the survey of various medicinal plants of the Poland that have been found associated in curing urinary diseases.
Although the urinary diseases are emerging as burning issues and a lot of research is currently going on in identifying safe chemotherapeutic agents from the herbal medicinal plants. This study could have been a milestone in encompassing all those plants and their chemical ingredients with their compressive details. Unfortunately, this manuscript offers limited information about the phytochemicals responsible for the activities. It only focuses only on the plants extracts which is contradictory to the title of this manuscript. The other major points of concern are as follows;
1. Apart from the overlap of the published preprint, the manuscript shows some overlap in the other parts; For instance, in biological protocols at Page-14, in all tables (Table-1-4). The overlap report is attached as a proof..
2. Interestingly the article available online with the title “Podkarpackie Voivodeship (Poland) To Treat Urinary Diseases” consisting of the list of the same plants of “Podkarpackie Voivodeship which is one of the provinces of Poland. How the authors in this current manuscript can claim these plants as the plants of the whole country (Poland). This manuscript lacks so many plants which are although endemic to Poland showing potential against urinary diseases but are missing in this manuscript. For instance, Cucumis melo L., Cucumis Sativus L., Clitoria ternatea and many others. Thus, the title of the manuscript is not aligned to its data.
3. The manuscript is only providing surface information about the plants’ potentials against urinary diseases. It is not providing the in-depth quantities analysis of these plants such as magnitude of the biological potentials. Although it has been mentioned at few places in the main text but not throughout. Readers are always interested to know how much potential each plant is showing.
4. It seems that the information provided in the tables has no relevance to the data provided in the main texts as we do not see any of the tables referred in the main text. The authors are encouraged to properly refer and discuss the data provided in the tables in the main text.
5. The conclusion part is not comprehensive and is not highlighting the overall events of this whole manuscript.
6. The title of first column of the Table-1 stating “Species full name by family” is a bit inappropriate and should be replaced by the title “Name of the species (Family)” and the family name should be in the bracket as a secondary information.
7. The words “infusion, decoction” in the fifth column of Table-1 (Utilization and administration) do not give any sense. The authors are encouraged to use proper format for this throughout this Table.
8. In Table-4 the authors have been mentioned names of some common edible plants. My concern is, what it has to do with the title of this manuscript. The authors are encouraged to properly mention or explain whether these plants are both edible as well as having potential against UTI or just edible ones. If so, it will be quite odd and should be deleted.
9. The manuscript is not properly arranged or formatted. There are many formatting mistakes. Extra spaces are given throughout. The authors are encouraged to properly format the whole manuscript.
10. Many of the references are out of the format of this journal. For instance, Bahmani et al. 2018. The biological names are not italic at many places in this manuscript especially those given in the references. For instance, Origanum vulgare, Hypericum perforatum, Fragaria vesca and so on.
This manuscript requires extensive corrections and modifications. I will encourage the authors to address all the highlighted issues and resubmit the manuscript for further consideration.

Comments on the Quality of English Language
There are many sentences in this manuscript which are not providing the required sense. The manuscript is required to be revisited throughout for the possible corrections and modifications of sentences.
Author Response
The manuscript entitled “New light on plants and their chemical compounds used in Polish folk medicine to treat urinary diseases” is review article based on the survey of various medicinal plants of the Poland that have been found associated in curing urinary diseases.
Although the urinary diseases are emerging as burning issues and a lot of research is currently going on in identifying safe chemotherapeutic agents from the herbal medicinal plants. This study could have been a milestone in encompassing all those plants and their chemical ingredients with their compressive details. Unfortunately, this manuscript offers limited information about the phytochemicals responsible for the activities. It only focuses only on the plants extracts which is contradictory to the title of this manuscript. The other major points of concern are as follows;
Response: We added more information about the phytochemicals responsible for the activities described in the manuscript and their mechanisms of action.
- Apart from the overlap of the published preprint, the manuscript shows some overlap in the other parts; For instance, in biological protocols at Page-14, in all tables (Table-1-4). The overlap report is attached as a proof.
Response: The overlap is with a preprint called “Plants Used in Podkarpackie Voivodeship (Poland) To Treat Urinary Diseases”. This preprint was published without our knowledge and permission, and was never meant to be available online. The number one source of overlap listed in the report is www.researchsquare.com, which also has the preprint. The report also showed some overlap with other sources in Table 3, so we fixed this issue.
- Interestingly the article available online with the title “Podkarpackie Voivodeship (Poland) To Treat Urinary Diseases” consisting of the list of the same plants of “Podkarpackie Voivodeship which is one of the provinces of Poland. How the authors in this current manuscript can claim these plants as the plants of the whole country (Poland). This manuscript lacks so many plants which are although endemic to Poland showing potential against urinary diseases but are missing in this manuscript. For instance, Cucumis melo, Cucumis Sativus L., Clitoria ternatea and many others. Thus, the title of the manuscript is not aligned to its data.
Response: the „Podkarpackie Voivodeship” manuscript was mistakenly published as a preprint without our knowledge and consent, and was not meant to be available online. The current article describes plants used in Polish traditional medicine. We added more information about the selection of plants to the methodology section of the introduction.
- The manuscript is only providing surface information about the plants’ potentials against urinary diseases. It is not providing the in-depth quantities analysis of these plants such as magnitude of the biological potentials. Although it has been mentioned at few places in the main text but not throughout. Readers are always interested to know how much potential each plant is showing.
Response: This information has been added throughout the text:
- millefolium: “For example, hydroethanolic extract at the dose of 300 mg/kg increased diuresis by approximately 30-60% between 4 and 8 hours after administration.”
- calamus: “At the dose of 750 mg/kg the diuretic effect was comparable to that of furosemide. The urine volume increased from 1.88 ± 0.18 ml (control) to 3.78 ± 0.11 ml (A. calamus extract at the dose of 750 mg/kg) and 4.02 ± 0.17 ml (furosemide at the dose of 15 mg/kg).”
Apium graveolens: “Al Jawad et al. (2011) observed that celery (at the dose of 8 g/kg/day) increased urinary Ca2+ excretions (from 2.05 ± 0.07 to 5.01±0.05 mmol/L) in an experimental model of nephrocalcinosis, and caused a significant reduction of serum creatinine (from 97.3 ± 0.5 to 87.2±0.63 mmol/L) and blood urea nitrogen (from 7.3 ± 0.2 to 5.7 ± 0.5 mmol/L).”
Daucus carota: “Both used doses (200 and 400 mg/kg) had significant effects. The dose of 400 mg/kg decreased the levels of serum urea (from 79.10 ± 2.21 to 45.96 ± 2.37 mg/dl), blood urea nitrogen (from 37.23 ± 1.36 to 20.21 ± 3.38 mg/dl), uric acid (from 4.87 ± 0.29 to 2.83 ± 0.35 mg/dl), and creatinine (from 2.03 ± 0.08 to 1.04 ± 0.08 mg/dl).”
- It seems that the information provided in the tables has no relevance to the data provided in the main texts as we do not see any of the tables referred in the main text. The authors are encouraged to properly refer and discuss the data provided in the tables in the main text.
Response: We added this information:
“Table 1 presents the plant nomenclature, the main Polish name, the English common name, the part used, utilization, and administration. Most often, dried material was used. Moreover, in Table 1, plants are listed alphabetically by plant family, and then by species within them. The greatest numbers of plant species used to treat urinary diseases were found to come from the Compositae (10 species), Rosaceae (9 species), Apiaceae (7 species), and Leguminosae (6 species).”
“Table 2 presents the results of the researched studies, including the experimental model, extract used, dose, control and reference substance, duration, number of animals or participants, and effect on UV and UNa. The remaining test results are described in the text for individual plants or are included in the table in the ‘other effects found’ column: in this way, the text complements the information in the table rather than duplicates it.”
“Table 3 summarizes the most commonly used plants in Polish folk medicine for the treatment of urinary diseases, and their main phytochemicals. Flavonoids were found in every plant species listed in the table. Phenolic acids, coumarins, tannins, and terpenes were also common.”
- The conclusion part is not comprehensive and is not highlighting the overall events of this whole manuscript.
Response: We have rewritten the conclusion.
- The title of first column of the Table-1 stating “Species full name by family” is a bit inappropriate and should be replaced by the title “Name of the species (Family)” and the family name should be in the bracket as a secondary information.
Response: We corrected this.
- The words “infusion, decoction” in the fifth column of Table-1 (Utilization and administration) do not give any sense. The authors are encouraged to use proper format for this throughout this Table.
Response: We modified the table to make it clearer.
- In Table-4 the authors have been mentioned names of some common edible plants. My concern is, what it has to do with the title of this manuscript. The authors are encouraged to properly mention or explain whether these plants are both edible as well as having potential against UTI or just edible ones. If so, it will be quite odd and should be deleted.
Response: We deleted this.
- The manuscript is not properly arranged or formatted. There are many formatting mistakes. Extra spaces are given throughout. The authors are encouraged to properly format the whole manuscript.
Response: We formatted the manuscript.
- Many of the references are out of the format of this journal. For instance, Bahmani et al. 2018. The biological names are not italic at many places in this manuscript especially those given in the references. For instance, Origanum vulgare, Hypericum perforatum, Fragaria vesca and so on.
Response: We corrected the references.
Reviewer 4 Report
Comments and Suggestions for Authors
The article is interesting, however some recommendations are made.
The authors may consider organizing the plant section in a different way, perhaps organizing them better in terms of their use or treatment in urinary diseases, because there are plant sections with very little information.
In the phytochemical part, plant components are mentioned but some mechanisms are mentioned very superficially. It is necessary to discuss the structure-activity relationship, factors that influence the biological activity described. The authors could add images or figures indicating the different mechanisms of action of the compounds with respect to urinary infection, etc.
Table 4 would recommend separating the individual plants and indicating their specific use (spice, soups, juices, etc.) or eliminate it.
Figure 1 is missing. That's the description of the figure only.
Author Response
The article is interesting, however some recommendations are made.
Response: Thank you for reviewing the manuscript and for providing such helpful comments. All of them have been taken into consideration when revising the manuscript.
The authors may consider organizing the plant section in a different way, perhaps organizing them better in terms of their use or treatment in urinary diseases, because there are plant sections with very little information.
Response: We feel that the current method of organization makes it easier to distinguish the plants used in Polish folk medicine to treat urinary diseases, which is the focus of this manuscript; however, we added a new Figure (Figure 1) to summarize section 2 and make it easier to quickly check the activity of each plant.
In the phytochemical part, plant components are mentioned but some mechanisms are mentioned very superficially. It is necessary to discuss the structure-activity relationship, factors that influence the biological activity described. The authors could add images or figures indicating the different mechanisms of action of the compounds with respect to urinary infection, etc.
Response: We added more information about this:
“Arbutin is a phenolic glycoside that shows antimicrobial and anti-inflammatory activity. It can decrease the production of inflammatory cytokines, and reduce the expression of inducible NO synthase (iNOS). About 65% of arbutin undergoes hydrolyzation to hydroquinone, which takes place primarily in the intestines. The main mechanism of the antimicrobial activity of hydroquinone is tied to the destruction of the bacterial cell wall (Cela-López et al., 2021).” Rutin is a flavonoid that can be found in several plant species used in the treatment of urinary diseases, including Fraxinus excelsior, Matricaria chamomilla, or Viola tricolor (Barnes J. et al., 2007; Kostova & Iossifova, 2007; Rimkiene et al., 2003). It can decrease the levels of oxalate and calcium in kidneys and urine. Oxalate and calcium are the main components of urinary stones. This effect is thought to be due to the inhibition of of oxalate synthesis and the increase of calcium sequestration by nitric oxide (Pawar et al., 2010; Zeng et al., 2019). Moreover, Kappel et al. (2013) have shown, that rutin increases the uptake of calcium into skeletal muscles, which is mediated through mitogen-activated kinase (MEK) and protein kinase A (PKA) signalling pathways.
There are many different mechanisms of action of diuretic or aquaretic drugs. Based on their site of action, diuretics can be classified into thiazide (inhibition of the Na+/Cl- transporter), loop (inhibiton of the Na+/K+/2Cl-), potassium-sparing diuretics, and carbonic anhydrase inhibitors. There are also aquaretics that block the V2 vasopressin receptor reducing aquaporin 2 (AQP2) water channel and sodium/glucose transporter 2 (SGLT2) inhibitors. New mechanisms of action of diuretic compounds, for example inhibition of adenosine A1 receptors or urea transporters, are also being discovered (Titko et al., 2020).
Luteolin (at a dose of 3 mg/kg) showed diuretic activity in normotensive and spontaneously hypertensive rats. Moreover, in normotensive rats it did not change the calcium levels in urine, which is an important aspect of urinary stones formation prevention. Importantly, luteolin did not increase potassium excretion unlike thiazide and loop diuretics. The authors suggest that its mechanism of action might be linked to muscarinic acetylcholine receptors (Boeing et al., 2017). Similarly to luteolin, gallic acid also increased diuresis without increasing potassium excretion in rats at the dose of 3 mg/kg. The effect was similar to that of hydrochlorothiazide (a clinical reference thiazide-type diuretic). However, here, the effect was not dependant on the activation of muscarinic acetylcholine receptors. Instead, the mechanism of action of gallic acid might be linked to endogenous prostanoids generation (Schlickmann et al., 2018).”
Table 4 would recommend separating the individual plants and indicating their specific use (spice, soups, juices, etc.) or eliminate it.
Response: We deleted Table 4.
Figure 1 is missing. That's the description of the figure only.
Response: We added new Figure 1 (as we added another Figure earlier in the manuscript, Figure 1 is now Figure 2).
Round 2
Reviewer 1 Report
Comments and Suggestions for Authors
The revised manuscript submitted for publication in its current form is not acceptable for publication in esteemed "Pharmaceuticals” journal. Although authors have addressed each queries very well rose by reviewer and have critically modify the manuscript as per requirement but manuscript requires some minor revision in the revised manuscript. Hence, it is requested that authors should revised the manuscript according to following suggestions.
Minor Comments:
1. In abstract, authors should incorporate the conclusions of finding reported in the manuscript.
2. It is suggested that authors can discuss the mechanism of action about role of plant’s extract in the treatment urinary diseases with suitable figure for better understanding.
3. Text alignment in Table 2 is not appropriate. Please realign the text to make it clearer.
4. Caption of Figure 2 should be explanatory. Please define the abbreviation used in Figure 2.
Author Response
The revised manuscript submitted for publication in its current form is not acceptable for publication in esteemed "Pharmaceuticals” journal. Although authors have addressed each queries very well rose by reviewer and have critically modify the manuscript as per requirement but manuscript requires some minor revision in the revised manuscript. Hence, it is requested that authors should revised the manuscript according to following suggestions.
Response: Thank you for reviewing the manuscript and for providing such helpful comments. All of them have been taken into consideration when revising the manuscript.
Minor Comments:
In abstract, authors should incorporate the conclusions of finding reported in the manuscript.
Response: We have added more information about it in the Abstract: “Both in vitro and in vivo studies have confirmed that many of these plant species have beneficial properties, such as diuretic, antihyperuricemic, antimicrobial, and anti-inflammatory activity, or prevention of urinary stones formation. These effects are exerted through different mechanisms, for example through the activation of bradykinin B2 receptors, inhibition of xanthine oxidase, or inhibition of Na+-K+ pump. Many plants used in folk medicine are rich in phytochemicals with proven effectiveness against urinary tract diseases, such as rutin, arbutin, or triterpene saponins.”
- It is suggested that authors can discuss the mechanism of action about role of plant’s extract in the treatment urinary diseases with suitable figure for better understanding.
Response: We have added a new figure (Figure 2).
- Text alignment in Table 2 is not appropriate. Please realign the text to make it clearer.
Response: We have corrected the text alignment of Table 2.
- Caption of Figure 2 should be explanatory. Please define the abbreviation used in Figure 2.
Response: We have defined the abbreviations in Figure 2 (now renamed to Figure 3).
Reviewer 2 Report
Comments and Suggestions for Authors
Dear authors,
I appreciate that the authors made many improvements in the manuscript. But additions/modifications are still needed, respectively:
There are still typos in the text (e.g. ml to be corrected in mL)
I recommend removing references older than 10 years. In addition, being a review, it is recommended to create a flow prism that helps to appreciate the references used.
Author Response
I appreciate that the authors made many improvements in the manuscript. But additions/modifications are still needed, respectively:
There are still typos in the text (e.g. ml to be corrected in mL)
I recommend removing references older than 10 years. In addition, being a review, it is recommended to create a flow prism that helps to appreciate the references used.
Response: Thank you for reviewing the manuscript and for providing such helpful comments. All of them have been taken into consideration when revising the manuscript.
We have corrected. Now, it is “mL”.
Because this was not intended to be a systematic review, we did not collect data needed for prisma flow diagram. However, we added more information about our search method. It is true that some references in this article are older than 10 years, but we feel that they are still relevant to this study, as they discuss the use of different plants in Polish folk medicine. Moreover, there are above 50 articles, which were published between 2013-2024.
Reviewer 3 Report
Comments and Suggestions for Authors
The revised version of the manuscript entitled “New light on plants and their chemical compounds used in Polish folk medicine to treat urinary diseases” has been found in much more improved condition. The authors have incorporated all the required changes in this whole manuscript apart some minor issues in the Table-1. The authors are encouraged to use the same format of writing the botanical names and their respective family name as "Botanical name (Family name) throughout Table-1.
I recommend this manuscript for publication in this journal after incorporating the above minor changes. The editor is requested to confirm that the required changes have been made accordingly before this manuscript is being processed onward.
Author Response
The revised version of the manuscript entitled “New light on plants and their chemical compounds used in Polish folk medicine to treat urinary diseases” has been found in much more improved condition. The authors have incorporated all the required changes in this whole manuscript apart some minor issues in the Table-1. The authors are encouraged to use the same format of writing the botanical names and their respective family name as "Botanical name (Family name) throughout Table-1.
I recommend this manuscript for publication in this journal after incorporating the above minor changes. The editor is requested to confirm that the required changes have been made accordingly before this manuscript is being processed onward.
Response: We have corrected this.
Reviewer 4 Report
Comments and Suggestions for Authors
The authors complemented and organized the article in a better way. They considered all the recommendations made.
Author Response
The authors complemented and organized the article in a better way. They considered all the recommendations made.
Response: Thank you for reviewing the manuscript.